# Do Land Use Changes Balance out Sediment Yields under Climate Change Predictions on the Sub-Basin Scale? The Carpathian Basin as an Example

**Paulina Orlińska-Woźniak** [1],* , **Ewa Szalińska** [2] **and Paweł Wilk** [1]

[1] Institute of Meteorology and Water Management—National Research Institute, 01-673 Warsaw, Poland; pawel.wilk@imgw.pl

[2] Departament of Environmetal Protection, Faculty of Geology, Geophysics and Environmental Protection, AGH University of Science and Technology, A. Mickiewicza Ave. 30, 30-059 Cracow, Poland; eszalinska@agh.edu.pl

\* Correspondence: paulina.wozniak@imgw.pl

**Abstract:** The issue of whether land use changes will balance out sediment yields induced by climate predictions was assessed for a Carpathian basin (Raba River, Poland). This discussion was based on the Macromodel DNS (Discharge–Nutrient–Sea)/SWAT (Soil and Water Assessment Tool) results for the RCP 4.5 and RCP 8.5 scenarios and LU predictions. To track sediment yield responses on the sub-basin level the studied area was divided into 36 units. The response of individual sub-basins to climate scenarios created a mosaic of negative and positive sediment yield changes in comparison to the baseline scenario. Then, overlapped forest and agricultural areas change indicated those sub-basins where sediment yields could be balanced out or not. The model revealed that sediment yields could be altered even by 49% in the selected upper sub-basins during the spring-summer months, while for the lower sub-basins the predicted changes will be less effective (3% on average). Moreover, the winter period, which needs to be re-defined due to an exceptional occurrence of frost and snow cover protecting soils against erosion, will significantly alter the soil particle transfer among the seasons. Finally, it has been shown that modeling of sediment transport, based on averaged meteorological values and LU changes, can lead to significant errors.

**Keywords:** sediment yield; land use change; climate change; sub-basins; Macromodel DNS/SWAT

## 1. Introduction

Land use (LU) and atmospheric factors, such as precipitation and temperature, exert a huge impact on the amount of sediment yield on practically every basin. Studies conducted in many regions of the world [1–7] have proven that surface runoff is a serious threat to soil resources in the world [8,9]. This phenomenon is particularly dangerous in arable lands because the maintenance of long-term crop production depends on the soil's production capacity, which is negatively affected by leaching of the topsoil and organic matter, and increased water outflow [10–13]. Until recently, in temperate zones, frost and snow cover were playing a similar role during winter as vegetation cover for the rest of the year. Nowadays, even in mountainous areas, frost and snow are becoming rarer and the erosive effect of rain is increasing during this period [14–16]. The problem is even more acute in sub-mountain and mountainous areas, where large slopes accelerate the leaching of soil particles. The correct identification of such areas is not simply easy because even relatively small river basins can display huge variability in terms of land and climate features that greatly affect sediment yield calculations [17,18]. Forecasts indicate that both climate and LU change will be very dynamic this century [19–22]. In turn, changes in LU will depend on local economics, population migration, arable land quality, and their location

relative to urban and protected areas, as well as national and international policies [23]. Both of these stressors will interact by summation or balancing, depending on the region, basin, or even sub-basin area. However, knowledge about their combined impact is still limited due to the small amount of comprehensive research [13,17,18,24–30]. The mountainous area, where the Raba River basin is located, is very specific in terms of land use forecasts, meteorological phenomena, intensity of water erosion [31], as well as activities to mitigate this process [32]. Until now, most similar studies have been conducted in catchments where the main problem was the development of cities/agriculture. Meanwhile, forecasts for the Raba basin are quite opposite. Forests are slowly beginning to dominate the landscape of this area [33–35], taking the place of agriculture, and the falling number of inhabitants contributes to a reduction of urban area impact on the environment, despite the proximity of a large urban agglomeration—the Krakow Metropolitan Area.

The goal of this study is to answer the question of whether LU changes will be able to compensate for sediment yield increase predicted by the future climate change scenarios on the individual sub-basin scale. The modeling tool used in this study, the Macromodel DNS (Discharge–Nutrient–Sea) combined with the SWAT module (Soil and Water Assessment Tool), enabled also the discussion on seasonal trends of this process.

## 2. Materials and Methods

### 2.1. Study Area Description

The Raba River basin has a surface of 151,700 ha and is located in the southern part of Poland in the area of the Polish Carpathian Mts. The approximately 132 km long river is divided into two parts (Figure 1) by an impoundment reservoir (60.1 km) which makes the main source for drinking water for approx. half a million people [36,37]. The relative parts of the basin are very different in terms of terrain slope, soil type, and land use (Figure 2a–c) with the mountainous part located upstream from the reservoir, and a sub-montane following the downstream reach of the river. Therefore, in this modeling approach the studied area was divided into two separate zones, i.e., the upper and lower Raba River basins. Since the entire Raba River basin is located in an area particularly vulnerable to water erosion [12,38–40] the issue of sediment runoff has been studied previously, especially in the direct basin of the reservoir [41–43]. However, no approach has been attempted to address this issue on the sub-basin level, except for sediment load prediction for the upper part of the basin [44].

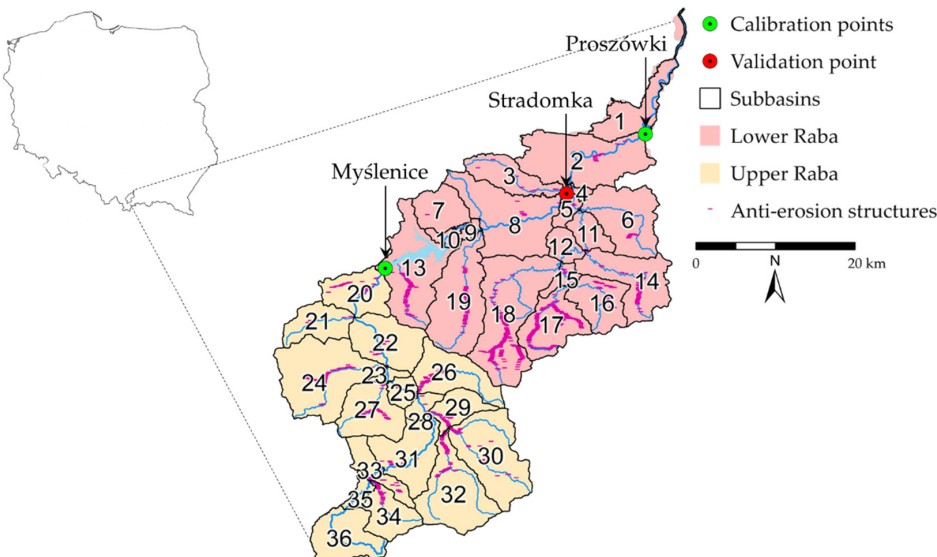

**Figure 1.** The Raba River basin localization along with division into upper and lower parts, sub-basins and, location of anti-erosion structures.

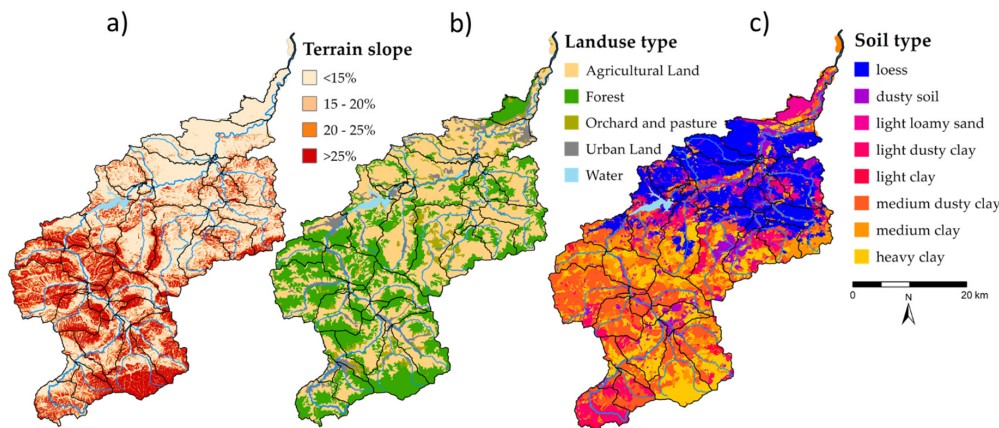

**Figure 2.** The Raba River basin along with: (**a**) terrain slope, (**b**) land use, and (**c**) type of soils arranged in terms of susceptibility to erosion.

The entire Raba River basin has been divided into 36 sub-basins (Figure 1) as described in Section 2.2. The upper Raba River consists of 17 sub-basins (No. 20–36) with the total area of nearly 74,900 ha. Almost 43% of this part is covered by slopes over 25% and overgrown by forest. The lower slopes are used mostly for agricultural activities (over 40% of area) (Figure 3). Only one sub-basin, No. 35, is covered mostly by urban land use. The lower Raba River part consists of 19 sub-basins (No. 1–19) with a total area of 76,800 ha. Since the majority of this area is covered with smaller slopes, agriculture dominates its cover (72%) with a small share of forest (17%). However, the presence of sub-basins with dominant one type of the land-use should be noted. For instance, No. 4 with 100% of its area covered by agriculture, or No. 2 and 10 where urban land use prevails (Figure 3).

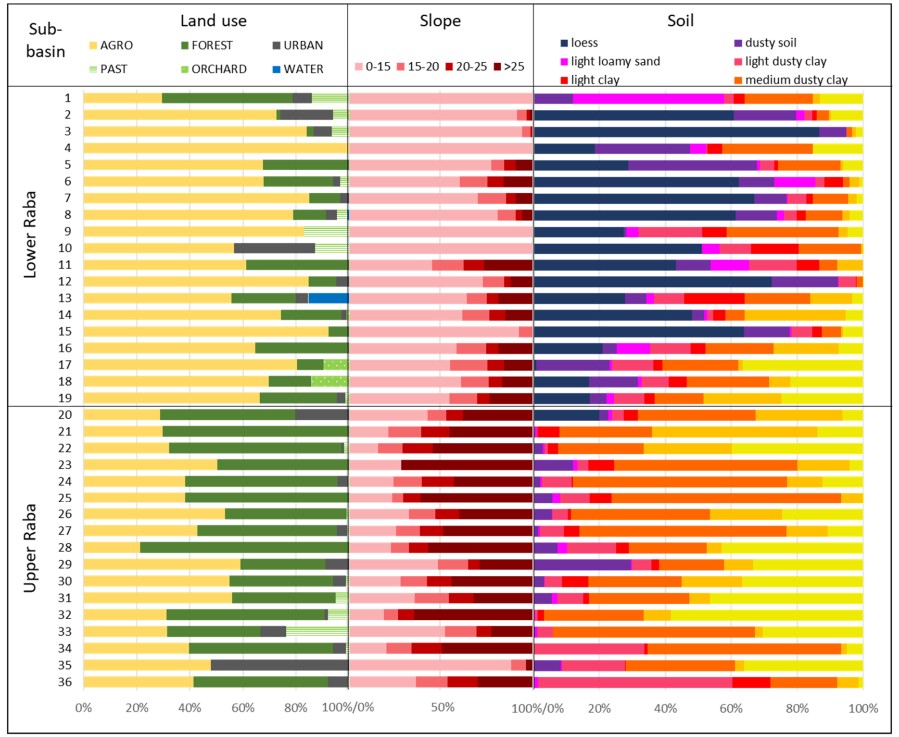

**Figure 3.** Area, land use, and slopes for each sub-basin.

## 2.2. Macromodel DNS/SWAT Description

To address changes in sediment yield provoked simultaneously by climate and LU scenarios in the studied basin the Macromodel DNS/SWAT, developed at the Institute of Meteorology and Water Management – National Research Institute (IMGW-PIB), was used [45–49]. Briefly, the Macromodel provides a platform where the SWAT module is integrated with other models (e.g., hydrological and meteorological), and in-situ/remote basin data. SWAT [50,51] is a physical model for continuous-time simulation using distributed-parameters developed to evaluate the effects of land use and alternative management decisions on water resources and point- and non-point-source pollution in river basins [50]. It is a watershed-scale deterministic model that operates at a daily time step using Digital Elevation Model (DEM), soil properties, LU information, and climate data as major inputs. The basin partition based on the hydrographic network, was further sub-divided into hydrological response units (HRUs), composed of unique combinations of land use, soil type, and slope [52]. Therefore, land use changes for variant scenarios were applied using a SWAT model (version 2012) functionality which allows for dynamic changes during simulation imitating the surface changes for individual types of land use [53]. To analyze the surface runoff in each sub-basin, the MUSLE (Modified Universal Soil Loss Equation) module [54,55] was used, which is a modified version of the Universal Soil Loss Equation (USLE) [56]:

$$sed = 11.8 \times \left( Q_{surf} \times q_{peak} \times area_{HRU} \right)^{0.56} \times K_{USLE} \times C_{USLE} \times P_{USLE} \times LS_{USLE} \times CFRG \tag{1}$$

where: *sed* is the sediment yield on a given day (metric tons), $Q_{surf}$ is the surface runoff volume (mm $H_2O$/ha), $q_{peak}$ is the peak runoff rate ($m^3$/s), $area_{HRU}$ is the area of the HRU (ha), $K_{USLE}$ is the USLE soil erodibility factor (0.013 metric ton $m^2$hr/($m^3$-metric ton cm)), $C_{USLE}$ is the USLE cover and management factor, $P_{USLE}$ is the USLE support practice factor, $LS_{USLE}$ is the USLE topographic factor, *CFRG* is the coarse fragment factor.

### 2.2.1. Model Set Up and Simulation

A SWAT model for the Raba River basin was set up and parameterized using the GIS interface of SWAT (ArcSWAT). The following data was used to build the Macromodel DNS/SWAT:

- map of Poland hydrographical divisions, scale of 1:10,000 (source: IMGW-PIB, resolution: 5 m);
- digital elevation model (DEM), scale of 1:20,000 (source: IMGW-PIB, resolution: 10 m);
- land use map—based on Corine Land Cover (CLC 2012), and agrotechnical data from the Local Data Bank (Figure 2b) (source: Copernicus Programme, resolution 20 m);
- soil map—detailed data on soil types, scale of 1:5000 (Figure 2c) (source: Institute of Soil Science and Plant Cultivation, resolution 2.5 m);
- meteorological data (1992–2016, e.g., precipitation and temperature) for 75 stations located directly in the basin, and within 20 km from its borders (source: IMGW-PIB);
- surface water quality data for suspended sediment (source: Polish State Monitoring System).

Hydrological and meteorological data have been introduced for the period of 27 years (1991–2017). However, due to the fact that data from sewage treatment plants and surface water quality in Poland have been available since 2005, calibration and validation of the model including sediment could have been carried out for a period of 13 years (2005–2017). The first 2 years of the simulation were used to condition the model [52].

### 2.2.2. Model Calibration/Validation

Calibration, verification, and validation of the model were performed using the SWAT-CUP program developed by [57]. In this study, the SUFI-2 algorithm was used to investigate sensitivity and uncertainty in streamflow prediction. Sensitivity analysis performed with the Latin Hypercube One-factor-at-a-Time (LH-OAT) sampling approach [58–60] was used to identify the most influential

model parameters for simulating the observed data. It gives two types of results, the value of statistics "*t*", and the level of significance "*p*". The smaller the value of "*p*", the more sensitive the parameter. In turn, the value of "*t*" indicates the intensity and direction of change of a given parameter (positive values mean its increase and negative values a decrease) (Table 1).

**Table 1.** Calibration of Soil and Water Assessment Tool (SWAT) parameters for the upper and lower Raba River Basin sorted by t-statistics.

| Parameter Name | Definition | t-Stat | *p*-Value |
|---|---|---|---|
| | Upper Raba | | |
| SURLAG.hru | Surface runoff lag coefficient | −1.04 | 0.30 |
| USLE_K(1).sol | USLE equation soil erodibility (K) factor | −0.70 | 0.48 |
| SOL_K(1).sol | Saturated hydraulic conductivity | −0.45 | 0.66 |
| PRF_BSN.bsn | Peak rate adjustment factor for sediment routing in the main channel | −0.40 | 0.69 |
| CH_K2.rte | Effective hydraulic conductivity in the main channel alluvium | −0.31 | 0.76 |
| ESCO.hru | Soil evaporation compensation factor | −0.25 | 0.81 |
| SPEXP.bsn | Exponent parameter for calculating sediment reentrained in channel sediment routing | 0.04 | 0.97 |
| CH_COV1.rte | Channel erodibility factor | 0.11 | 0.91 |
| CH_COV2.rte | Channel cover factor | 0.15 | 0.88 |
| ADJ_PKR.bsn | Peak rate adjustment factor for sediment routing in the subbasin | 0.81 | 0.42 |
| SPCON.bsn | Linear parameter for calculating the maximum amount of sediment that can be reentrained during channel sediment routing. | 0.89 | 0.37 |
| SOL_AWC(1).sol | Available water capacity of the soil layer | 1.37 | 0.17 |
| CH_N2.rte | Manning's "n" value for the main channel | 5.51 | 0.00 |
| USLE_P.mgt | USLE equation support practice | 7.49 | 0.00 |
| CN2.mgt | Initial SCS runoff curve number for moisture condition | 16.20 | 0.00 |
| HRU_SLP.hru | Average slope steepness | 20.80 | 0.00 |
| | Lower Raba | | |
| GW_DELAY.gw | Groundwater delay time | −1.47 | 0.14 |
| USLE_P.mgt | USLE equation support practice | −1.17 | 0.24 |
| SURLAG.hru | Surface runoff lag coefficient | −1.02 | 0.31 |
| USLE_K(1).sol | USLE equation soil erodibility (K) factor | −0.32 | 0.75 |
| SPEXP.bsn | Exponent parameter for calculating sediment reentrained in channel sediment routing | 0.04 | 0.97 |
| CH_COV2.rte | Channel cover factor | 0.08 | 0.94 |
| RES_SED.res | Initial sediment concentration in the reservoir | 0.62 | 0.54 |
| CN2.mgt | Initial SCS runoff curve number for moisture condition | 0.87 | 0.39 |
| SPCON.bsn | Linear parameter for calculating the maximum amount of sediment that can be reentrained during channel sediment routing | 0.89 | 0.37 |
| ADJ_PKR.bsn | Peak rate adjustment factor for sediment routing in the subbasin | 1.08 | 0.28 |
| CH_COV1.rte | Channel erodibility factor | 1.17 | 0.24 |
| RES_RR.res | average daily principal spillway release | 1.17 | 0.24 |
| PRF_BSN.bsn | Peak rate adjustment factor for sediment routing in the main channel | 1.46 | 0.15 |
| ALPHA_BF.gw | Baseflow alpha factor | 1.62 | 0.11 |
| RES_NSED.res | Normal sediment concentration in the reservoir | 2.42 | 0.02 |
| HRU_SLP.hru | Average slope steepness | 5.84 | 0.00 |

Model calibration and validation were performed with use of the flow data obtained from the IMGW-PIB, and total suspended sediment concentrations from the state monitoring system. Due to the fact that the state monitoring frequency is relatively low (12 times per year) the LOAD ESTimator program (LOADEST) [61] was used to assist in the development of the regression model for reliable

estimation of constituent load (calibration). Explanatory variables within the regression model included functions of streamflow, decimal time, and additional user-specified data variables. The formulated regression model then was used to estimate loads over a user-specified time interval (estimation). Mean load estimates, standard errors, and 95 percent confidence intervals, were developed on a monthly and seasonal basis [62,63]. Due to the dual character of the Raba River basin, described in Section 2.1, the calibration process was performed for the two parts of the basin separately using various parameters, ordered by sensitivity, described in Table 1.

For the upper part, the basin calibration was performed in the calculation profile of Myślenice, while the profile of Proszówki was selected for the lower part (Figure 1). These are the points closing the upper and lower Raba area for which monitoring data for flows and sediment were performed by Polish State Environmental Monitoring. The results obtained for both profiles were used to calibrate and verify the model. Validation of the model was performed for the Stradomka River, which is a right-bank tributary of the Raba River and is also subjected to suspended sediment state monitoring measurements (Figure 1). To evaluate the fit of the model to the monitoring results, four statistical measures were used, determination coefficient ($R^2$), efficiency coefficient of the Nash-Sutcliffe model (NSE), percent bias (PBIAS), and Kling-Gupta efficiency (KGE), which were described in detail [44,64–67] as well as appropriate ranges of values for these measures (Table 2). However, it should be remembered that, as recommended by the authors [68–70], these ranges of values should not be used rigidly, but in a flexible way, as there may be objective reasons that impede or prevent satisfactory assessment in some basins. These objective reasons include, among others, the specificity of the research area (described in more detail in Section 2.1) which translates into greater uncertainty in monitoring data constituting the basis for building the model.

**Table 2.** Classification of value ranges for statistical measures used during calibration, verification, and validation, based on: [68–70].

| Performance Rating | $R^2$ | NSE | PBIAS% | | KGE |
|---|---|---|---|---|---|
| | Flow/Sediments | Flow/Sediments | Sediments | Flow | Flow/Sediments |
| very good | >0.65 | $0.75 < \text{NSE} \leq 1$ | $<\pm25$ | $<\pm10$ | >0.75 |
| good | 0.5–0.65 | $0.5 < \text{NSE} \leq 0.75$ | $\leq\pm25 \text{ Pbias} < \pm40$ | $\leq\pm10 \text{ Pbias} < \pm15$ | 0.5–0.75 |
| satisfactory | 0.2–0.5 | $0 < \text{NSE} \leq 0.5$ | $\pm40 \leq \text{Pbias} < \pm70$ | $\pm15 \leq \text{P bias} < \pm25$ | 0–0.5 |
| nonsatisfactory | <0.2 | $\text{NSE} \leq 0$ | $\text{Pbias} \geq \pm70$ | $\text{Pbias} \geq \pm25$ | <0 |

Four statistical measures ($R^2$, NSE, PBIAS, and KGE) indicated model performance of Raba basin in flow simulation and sediment with a monthly time step for both used calculation profiles; Proszówki (Lower Raba), and Myślenice (Upper Raba). For flow calibration, statistical measures $R^2$ and KGE obtained values 0.62 and 0.73, and 0.70 and 0.80 (Table 3), respectively, which classify the model performance as good and very good. According to NSE, which obtained values 0.51 and 0.73, respectively, the performance of the model can be considered good (Table 3). Only PBIAS obtained 21% on the Proszówki profile, which, however, still qualifies it as satisfactory. For sediment calibration in the Myślenice calculation profile according to $R^2$ and NSE, which obtained values of 0.34 and 0.1, respectively, the performance of the model can be considered satisfactory. In turn, according to PBIAS and KGE, which obtained the value of respectively 2% and 0.58, the model's performance can be considered very good and good. Better results were obtained on the Proszówki calculation profile where, according to all four statistical measures, the performance of the model can be considered very good ($R^2$—0.77) and good (NSE—0.71, PBIAS—27% and KGE—0.69) (Table 3). To verify the model's prediction accuracy, it was also validated in the Stradomka calculation profile. For the flow, according to $R^2$, PBIAS and KGE, the model's performance can be considered good. Only NSE obtained a satisfactory value. In turn, for sediment, according to $R^2$, NSE, and KGE, the performance of the model can be considered satisfactory, and according to PBIAS as good (Table 3).

**Table 3.** The Raba River Model calibration and validation results for monthly simulations.

| Calculation Profile | Type | Interval | R$^2$ | NSE | PBIAS% | KGE |
|---|---|---|---|---|---|---|
| calibration | | | | | | |
| Myślenice | flow | 1993–2017 | 0.62 | 0.51 | 21 | 0.7 |
| | sediment | 2005–2017 | 0.34 | 0.1 | −2 | 0.58 |
| Proszówki | flow | 1993–2017 | 0.73 | 0.73 | 4 | 0.8 |
| | sediment | 2005–2017 | 0.77 | 0.71 | 27 | 0.69 |
| validation | | | | | | |
| Stradomka | flow | 1993–2017 | 0.57 | 0.46 | −14 | 0.72 |
| | sediment | 2005–2017 | 0.45 | 0.35 | 39 | 0.19 |

Due to the dual character of the analyzed basin, the statistical measures for the Myślenice calculation profiles displayed poorer model performance than for Proszówki (Table 3 and Figure 4). In the upper Raba River basin, the average annual rainfall amplitude is higher than in the lower part, which is associated with an altitude increase. In addition, short, but heavy rainfalls causing rapid flooding and anomalously wet seasons (AWS) are also more common in this part [71–74]. This precipitation often covers only a small part of the basin (e.g., one of the sub-basins). However, it is reflected in both flow and sediment yield simulations. Moreover, the low frequency of sediment monitoring by the Polish State Environmental Monitoring (SEM) makes it difficult to match the simulated sediment yield values with the values calculated from the monitoring data. The monitoring results of sediment are prone to errors, which is 15% of expanded uncertainty for analyzed measurements according to SEM. However, as reported by other studies [75], uncertainty can reach up to 70%, especially in mountainous areas. Such values affect, among the others, the obtained R2 values. Since this parameter is very sensitive to outliers, which are particularly common in case of sediment monitoring.

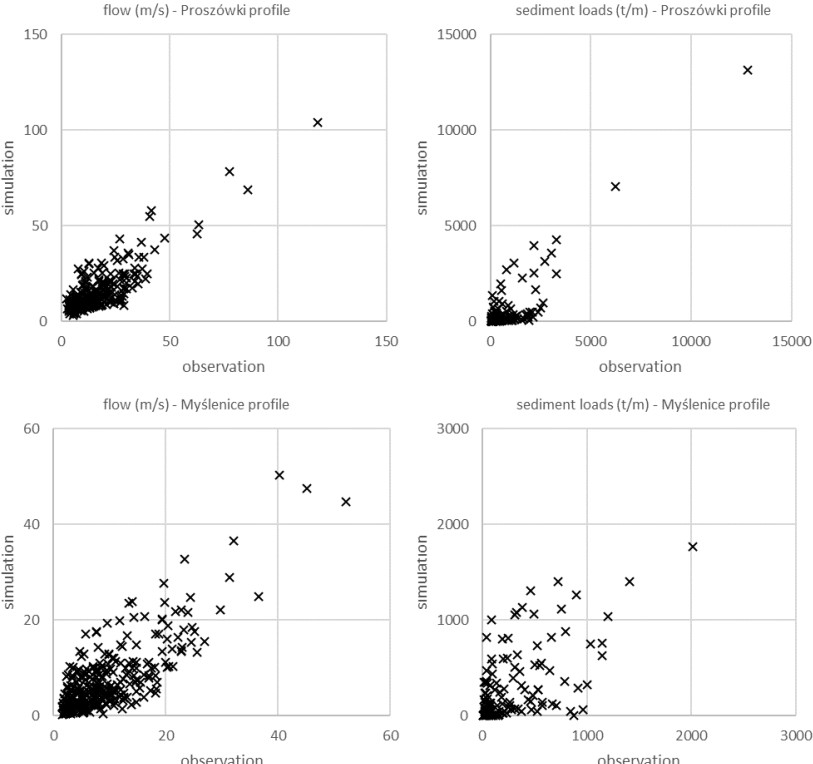

**Figure 4.** Calibration plots for flow and sediment loads.

It should also be remembered that statistical measures such as NSE are very sensitive to rapid water swells and criticized for many years for excessive sensitivity to high values [76], which explains why KGE was also used for analysis, as a more resistant parameter, displaying good and very good results for current analyzes. The calibrated and validated model has been adopted as the baseline scenario and served as the reference point for subsequent analysis.

*2.3. Scenarios*

In order to discuss the sediment yield issues at the sub-basin level two types of variants scenarios have been prepared:

- climate scenarios—taking into account the forecasted temperature and rainfall changes in the Raba River basin developed on the basis of RCP 4.5 and 8.5;
- land use scenarios (LU)—taking into account the forecast changes in land use of the Raba River basin (increase in forest and urban areas) developed as part of the FORECOM project [77].

2.3.1. Climate Scenarios

To account for predicted climate changes (precipitation and temperature) for two future time horizons (2021–2050 and 2071–2100) based on RCP 4.5 and RCP 8.5, the results of bias-corrected temperature and precipitation projections from an ensemble of EuroCORDEX regional climate models and CMIP5 general circulation models were used [78–80]. These changes will be unevenly distributed both in the country and individual catchments, and even individual parts of these drainage basins. However, high data resolution allowed for obtaining climate change forecasts for Myślenice (Upper Raba outflow profile) and Proszówki (the Lower Raba outflow profile). Thanks to this, different climate forecasts were separately introduced into the model for the sub-basins belonging to the Upper and Lower Raba [81]. These forecasts indicate that in the near (2021–2050) and long term (2071–2100), significant changes in average annual temperature and precipitation can be expected throughout the country. The amount of precipitation and temperature values strongly depend on the seasons. Therefore, the analyses were conducted with a monthly time step taking into account the division into seasons, winter (December, January, and February), spring (March, April, and May), summer (June, July, and August) and autumn (September, October, and November) (Figure 5). For the needs of the current study four climate scenarios for temperature and precipitation were prepared:

- C1.1—RCP 4.5 for the short-time perspective 2021–2050;
- C1.2—RCP 4.5 for the long-time perspective 2071–2100;
- C2.1—RCP 8.5 for the short-time perspective 2021–2050;
- C2.2—RCP 8.5 for the long-time perspective 2071–2100.

Under these scenarios, an increase of the average annual temperature by 1.2 °C could be expected both in the upper and lower Raba River basin parts. However, during the winter period it could reach 1.5 °C, and even 4.6 °C under the RCP 8.5 long-term predictions (Figure 5). For the remaining seasons, the temperature predictions did not display clear seasonal variability although such a pattern visible in the case of precipitation. The rainfall increase, even by 26% in the RCP 8.5 long-term forecasts, is to be expected during winter and spring in both parts of the basin. While the summer precipitation forecasts underline the difference between the upper and lower Raba basin parts, especially under the long-term forecasts. During this period precipitation in the upper Raba will decrease by 0.3–3%, while at the same time on the lower Raba precipitation an increase of even by 6% will be observed. In general, climate scenarios have shown that the winter and spring months will be characterized by the largest increases in precipitation and temperature, especially in the long term. Analysis of these forecasts indicates that the biggest changes will occur in winter. Especially in the case of long-term forecasts, which may lead to a complete disappearance of snow cover in the analyzed area.

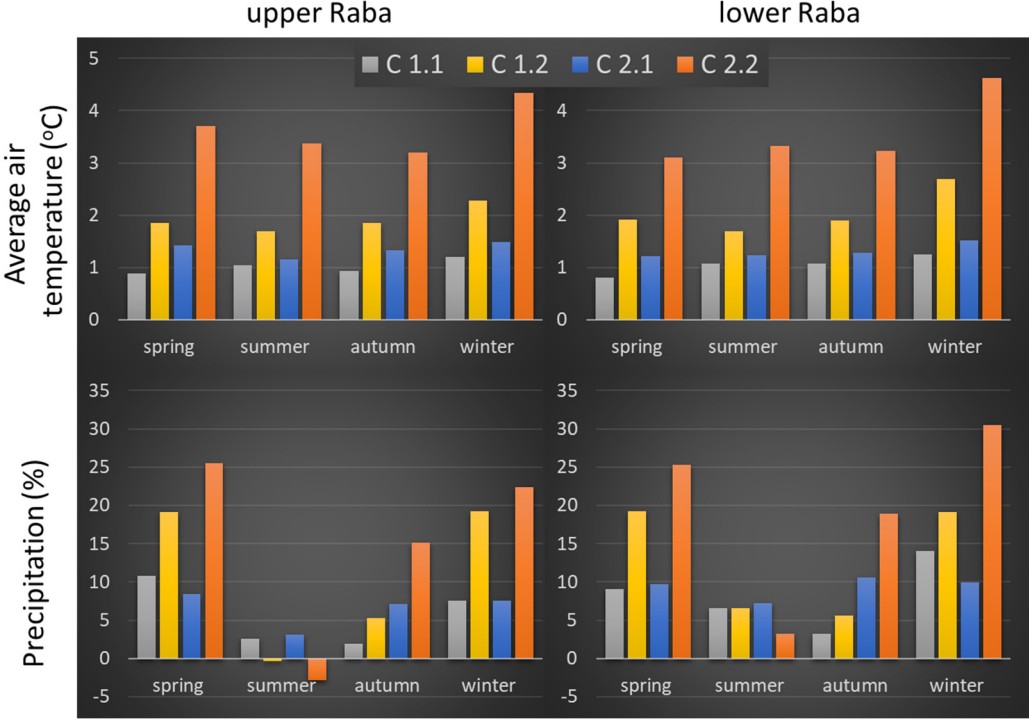

**Figure 5.** Temperature and precipitation changes for the upper and lower Raba River basins (Myślenice and Proszówki calculation profile) under RCP 4.5 and RCP 8.5 scenarios.

### 2.3.2. Land Use Scenarios (FOREST and URBAN)

Two LU scenarios were based on the FORECOM project, implemented to improve understanding of past, present, and future forest and urban cover changes in the Swiss Alps, and the Polish Carpathians in the context of climate changes. Two variant scenarios adopted in the current study were based on the DYNA-Clue model for the two Carpathian localities (Szczawnica and Niedźwiedź) for the future time horizon of 2060 [82]. The FORECOM project developed two forecasts:

- trend forecast—assuming the continuation of the dynamics of forest surface changes and land use established for the period of 1970–2013. In this forecast, forest and urban areas are projected to increase by 23% and 10%, respectively;
- liberal forecast—assuming that the directions of future land use changes will be primarily determined by free market mechanisms (with the main role played by the profitability of specific activities such as agriculture, forestry, or housing in the basin area). In this forecast, forest and urban areas are projected to increase by 30% and 15%, respectively.

Both forecasts assume that the growth of forest areas and areas of dispersed development will take place at the expense of areas that are currently used for agriculture.

The process of limiting agricultural land use in the studied area was started by post-war afforestation of former agricultural land and natural succession initiated at that time. The decrease in the area used as arable land results, as in the case of the development of urbanized areas, from socio-economic changes that have taken place since 1989. In the initial period of transformation, it was associated with the difficult economic situation of farmers, and problems resulting from changes in the profitability of production and the lack of demand for agricultural produce. In addition, EU structural programs supported the development of forest areas, which led to further abandonment of agricultural use. The fastest and most intensive afforestation process takes place on arable land located at higher land slopes where agricultural activity is difficult and soils are of poor quality [33]. The two forecasts (trend and liberal) from the FORECOM project used the starting point assigning the

catchment to the appropriate group and land slope size for each sub-basin. For the purposes of the FOREST variant scenario, two groups of slopes were created that defined the border between trend forecast and liberal forecast. Sub-basins for which land slopes fall within the range of <20% have been classified as group I. In this area, moderate growth of forest areas has been simulated in line with the trend forecast which means a 23% increase in forest areas compared to the baseline scenario. Sub-basins for which land slopes fall within the range of >20% were classified in group II. In this area, a more dynamic growth of forest areas was simulated in accordance with the liberal forecast, which means a 30% increase in forest areas compared to the baseline scenario (Figure 6). In both cases, the growing forest areas replaced the agricultural areas.

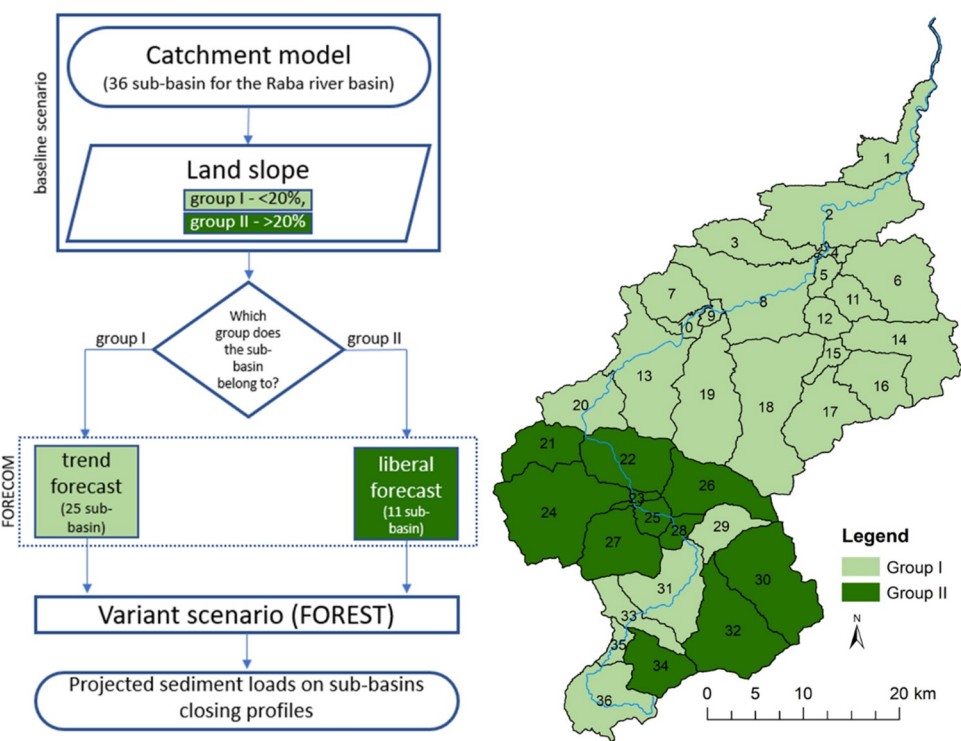

**Figure 6.** Procedure diagram for the sub-basin allocation to group I (trend forecast) or group II (liberal forecast) with their localization in the Raba River basin.

The URBAN variant scenario for the use of the catchment area allowed to simulate the forecasted increase in residential med/low density areas in the catchment area of the Raba River. The results obtained in the FORECOM project are consistent with the results of research and forecasts developed for both Poland and Europe, where a constant decrease in population has been observed for many years [83,84]. A large part of this area is still occupied by arable land, but over time, this type of land use is gradually losing importance and disappearing, giving way to residential med/low density areas which makes an attractive alternative to crowded city centers. Their development is determined by the growing Krakow Metropolitan Area (Figure 7). According to previous research (FORECOM) [33,82,85,86], urbanized and forested areas will be the two main types of land use that will replace agricultural areas in the next 50 years. This increase, however, will not be uniform throughout the catchment area. To account for these differences, the Statistics Poland—GUS forecasts for municipalities located in the Raba catchment area and two forecasts (trend forecast and liberal forecast) for urban areas developed in the FORECOM project were used to build the URBAN variant scenario. In order to assign the sub-basin to the appropriate FORECOM forecast, GUS data on future population migrations in the Małopolskie voivodeship were used [87]. The data for municipalities were then transferred to individual sub-basins using the proportion taking into account their area and

GUS data. It was assumed that the projected increase in the number of inhabitants would translate into an increase in residential med/low density areas.

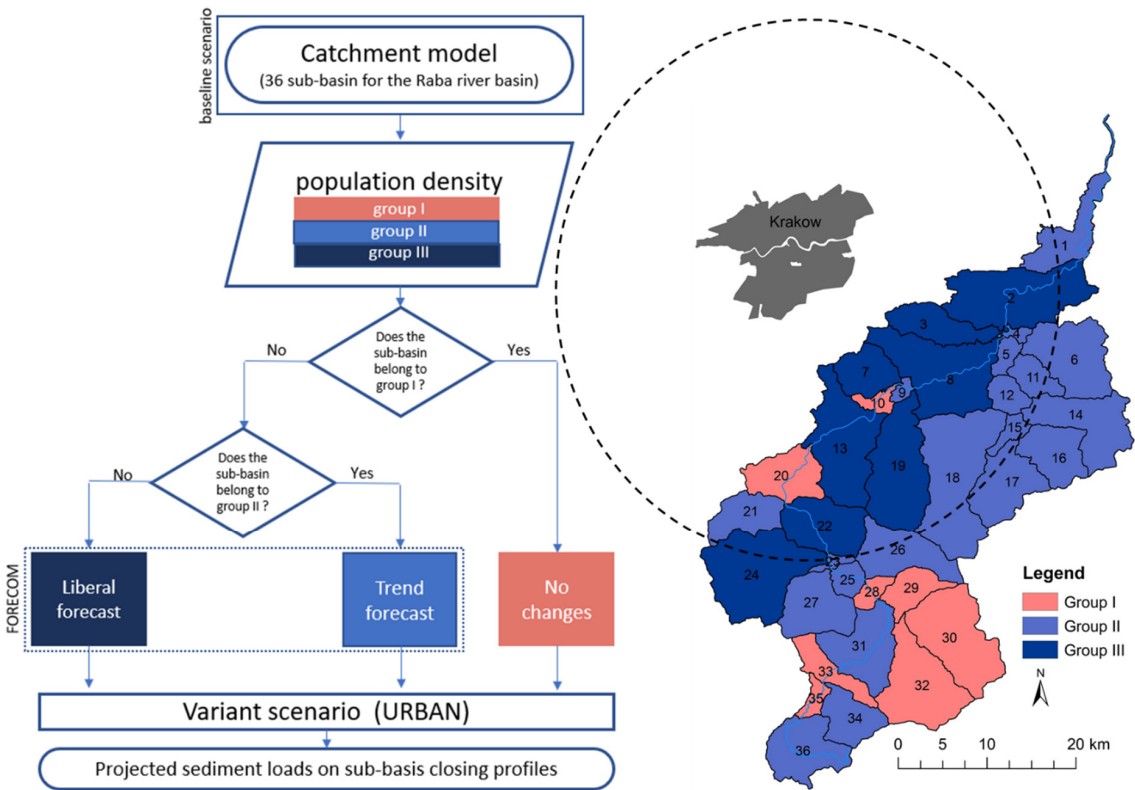

**Figure 7.** Procedure diagram for the sub-basin allocation to group I (no changes), group II (trend forecast), or group III (liberal forecast) with their localization in the Raba River basin and the extent of the Krakow Metropolitan Area.

All sub-basins were classified into three groups. The sub-basins for which a decrease of inhabitant number is forecasted were classified as group I (Figure 7). For these sub-basins no changes were made to the size of urban areas. Sub-districts on which a moderate increase of inhabitant number is forecasted have been qualified to group II. For sub-basins that were in this group, a trend forecast was used, i.e., an increase in residential med/low density areas by 10% compared to the baseline scenario. Sub-basins on which the largest increase of inhabitant number is forecasted have been qualified to group III. They are mostly sub-basins located within the reach of the Krakow Metropolitan Area. For sub-basins that are of this group, a liberal forecast was used, i.e., an increase in residential med/low density areas by 15% compared to the baseline scenario. The increase in residential med/low density areas in both groups II and III was at the expense of areas used for agriculture. The method of allocating sub-basins to the appropriate group is shown in Figure 7.

## 3. Results

The obtained Macromodel DNS/SWAT results enabled discussion of sediment yield temporal and spatial distribution in the Raba River basin. The detailed model response for each sub-basin for each of the scenario variables (temperature, precipitation, urban, and forest area predictions) and their combination have been included in the Mendeley Data [88]. As for the baseline scenario, considered as the reference point for subsequent examination of the climate and land use changes, important differences between seasons and sub-basins have been revealed (Figure 8, Table 4). Highest average sediment yields have been observed for the spring period, both in the upper and lower parts of the basin (0.92 +/− 0.27 t/ha and 0.57 +/− 0.34 t/ha, respectively). They were followed by the summer

sediment yield elevated values (0.58 +/− 0.30 t/ha and 0.41 +/− 0.23 t/ha, respectively) to reach minima during the autumn-winter period (at the range of 0.28–0.29 and 0.21–0.25, respectively). Although statistically significant differences (Kruskal-Wallis and Bonferroni tests, 95% confidence level) between the upper and lower parts of the Raba River basin have been detected only for the spring season, notably higher sediment yields in the upper part are also visible during the summer period. However, it should be noted that extremely high sediment yields have also been detected in one of the lower sub-catchments (No. 17).

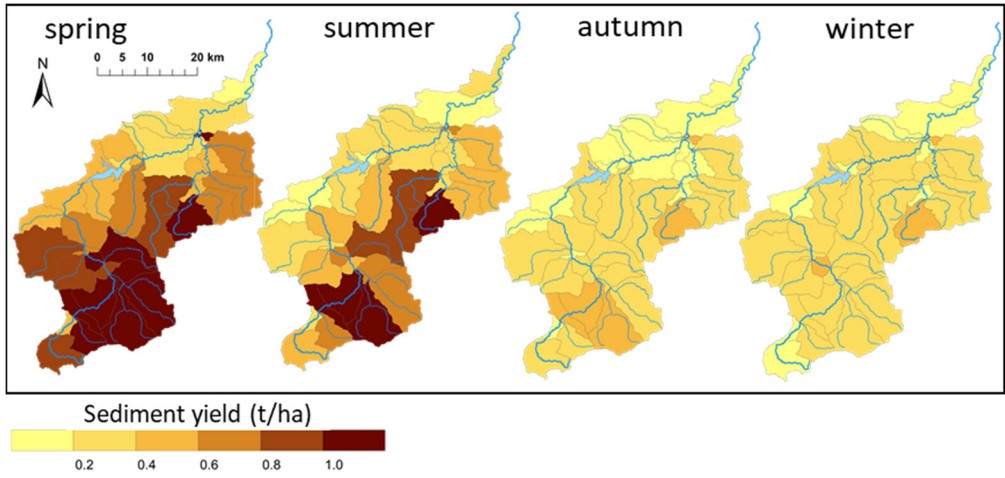

**Figure 8.** Baseline scenario for sediment yields for the Raba River individual sub-basins.

**Table 4.** Baseline scenario sediment yields (t/ha) for the upper and lower parts of the Raba River basin.

| | | Min | Sub-Basin No. | Max | Sub-Basin No. | Average | Standard Deviation |
|---|---|---|---|---|---|---|---|
| | | | | baseline scenario | | | |
| lower Raba | spring | 0.16 | 1 | 1.55 | 17 | 0.57 | 0.34 |
| | summer | 0.13 | 10 | 1.03 | 17 | 0.41 | 0.23 |
| | autumn | 0.08 | 10 | 0.53 | 4 | 0.21 | 0.12 |
| | winter | 0.09 | 1 | 0.6 | 4 | 0.25 | 0.12 |
| upper Raba | spring | 0.44 | 21 | 1.38 | 25 | 0.92 | 0.27 |
| | summer | 0.16 | 20 | 1.07 | 31 | 0.58 | 0.3 |
| | autumn | 0.11 | 35 | 0.48 | 31 | 0.28 | 0.12 |
| | winter | 0.11 | 35 | 0.43 | 25 | 0.29 | 0.08 |

The climate forecasts (Section 2.3.1) imposed on the baseline scenario further emphasized differences between seasons and sediment yield origins. The implications of the RCP 4.5 and RCP 8.5 predictions for the two-time perspectives (2021–2050 and 2071–2100) have been assessed for individual sub-basins as a sediment yield change in relation to the baseline calculations (Figure 9). The largest increases of the sediment yields have been observed in the upper part of the basin during the spring and winter periods. Particularly, in sub-basins No. 25, 30, and 32, where the predicted sediment yield is expected to grow by 0.15–0.30 t/ha (scenarios C1.1 and C2.2, respectively). The increase in sediment values has been also predicted for the selected lower sub-basins, especially during the winter period for No. 4 and 17 (0.21–0.26 t/ha). Sub-basin No. 17 stands out also during the summer period (scenario C1.1) displaying the biggest changes across the scenarios, from 0.21 t/ha in C1.1, to −0.18 t/ha in C2.2. Generally, the summer period is distinguished by a decrease of sediment yields, particularly noticeable in the C2.2 scenario, reaching values of −0.36 t/ha in sub-basin No. 25.

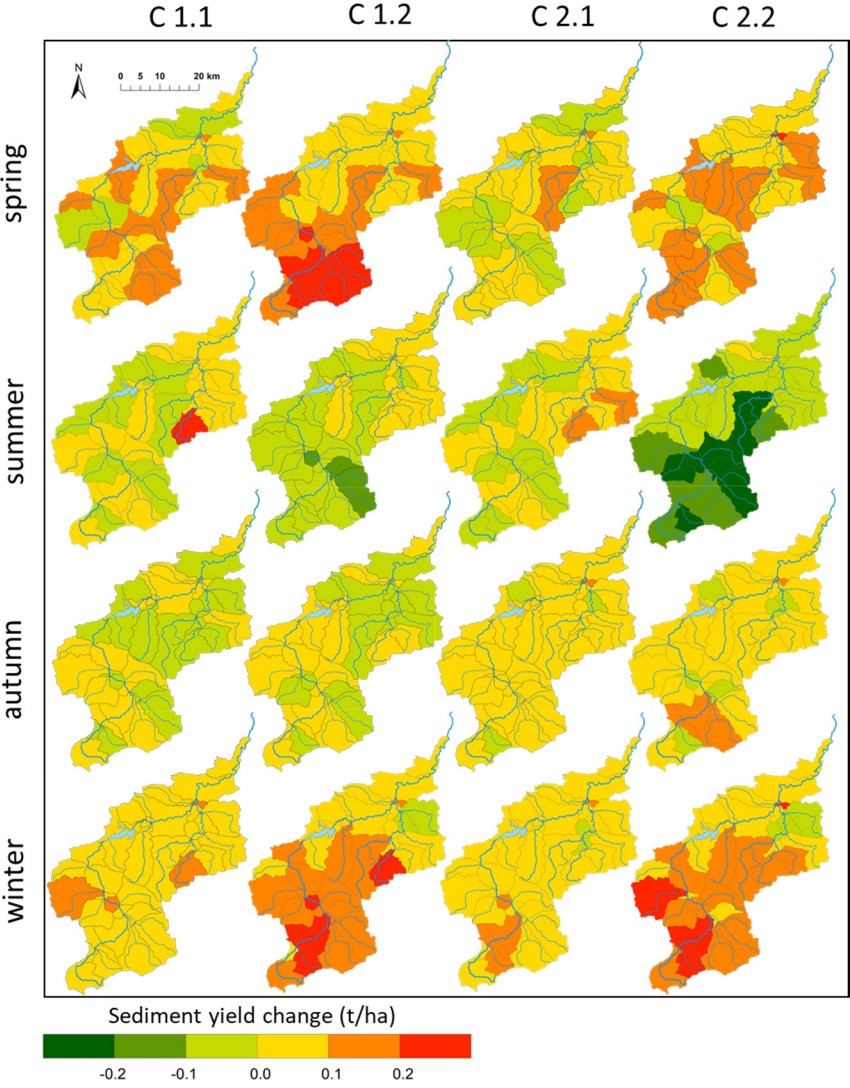

**Figure 9.** Sediment yield changes for individual climate scenarios compared to the baseline scenario.

As for the land use scenarios (LU, Section 2.3.2), a noticeable sediment yield reduction in the majority of the Raba River sub-basins has been observed when compared to the baseline scenario (Figure 10). Substantial average decreases have been observed in the upper part of the basin, especially in sub-basin No. 34, during the spring and summer periods (0.22 +/− 0.09 t/ha and 0.13 +/− 0.06 t/ha, respectively). As for the autumn and winter periods, the impact of LU scenarios on sediment yields was much smaller and remained below 0.1 t/ha. However, the presence of areas without any changes in sediment yield under LU scenarios should also be noted. Such a situation concerned sub-basins No. 35 (Upper Raba), and No. 4, 9, 10, and 15 (Lower Raba) characterized by a very small share of forest area, or its complete lack of.

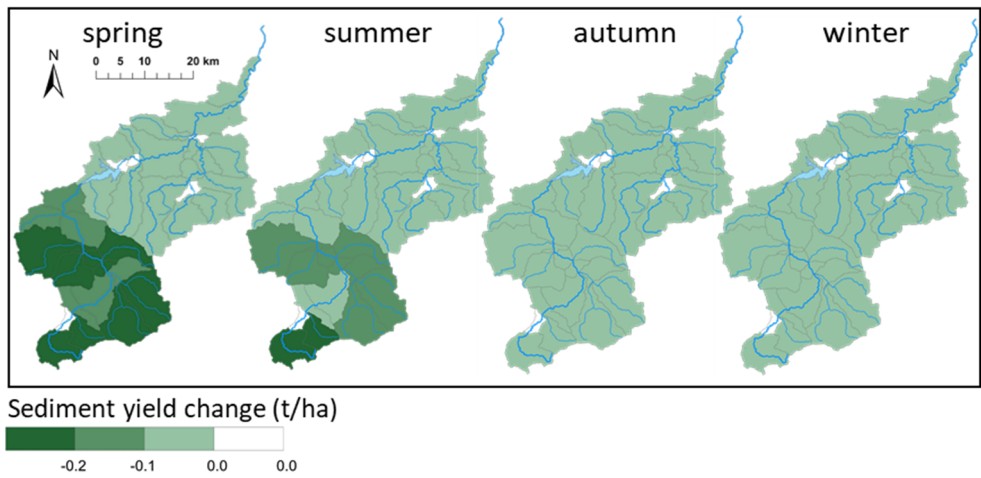

**Figure 10.** Sediment yield change under land use scenario for the individual sub-basins.

Since the assumed climate and land use changes are predicted to occur simultaneously, the final modeling approach focused on superimposition of previously described scenarios. As comparing seasonal average values for the upper and lower portions of the Raba River basin (Table 5) the general decrease of the sediment yields for the upper part has been noted in the range of 0.03–0.25 t/ha, except for the winter period under all scenarios. The scenario combination for the lower part resulted in a slight increase of sediment yields (0.01–0.06 t/ha) for the majority of the modeled cases. However, an average decrease by 0.01 has been also observed under RCP 4.5 scenarios (C1.1+LU and C1.2+LU) during the summer and autumn periods. Particularly, a large decrease of the seasonal average value in this part of the basin has been detected for the summer period under long term RCP 8.5 predictions (C2.2+LU).

**Table 5.** Sediment yield changes (t/ha) for the upper and lower parts of the Raba River basin under combination of climate change and land use scenarios.

|  |  | Average | Sd | Average | Sd | Average | Sd | Average | Sd |
|---|---|---|---|---|---|---|---|---|---|
|  |  | C 1.1 + LU | | C 1.2 + LU | | C 2.1 + LU | | C 2.2 + LU | |
| lower Raba | spring | 0.02 | 0.05 | 0.03 | 0.05 | −0.004 | 0.06 | 0.06 | 0.07 |
|  | summer | 0.01 | 0.05 | −0.01 | 0.05 | 0.01 | 0.04 | −0.08 | 0.07 |
|  | autumn | −0.01 | 0.02 | −0.01 | 0.02 | 0.01 | 0.03 | 0.01 | 0.04 |
|  | winter | 0.03 | 0.03 | 0.04 | 0.06 | 0.02 | 0.03 | 0.04 | 0.08 |
| upper Raba | spring | −0.16 | 0.1 | −0.1 | 0.12 | −0.21 | 0.11 | −0.15 | 0.11 |
|  | summer | −0.12 | 0.19 | −0.17 | 0.17 | −0.14 | 0.18 | −0.25 | 0.19 |
|  | autumn | −0.05 | 0.07 | −0.04 | 0.07 | −0.03 | 0.07 | 0.01 | 0.1 |
|  | winter | 0.01 | 0.03 | 0.09 | 0.05 | 0.01 | 0.03 | 0.07 | 0.06 |

To answer the main question of this study, the results obtained for the combined climate change and land use scenarios were compared with the baseline scenario for each sub-basin, taking into account the increase or decrease in sediment yield. The negative results (sediment yield change below zero; green color Figure 11) signify the ability of the LU scenario to compensate for the sediment yield changes induced by climate change. While, positive results (sediment yield change above zero; red color Figure 11) indicate a lack of such a response in individual sub-basins. During the spring season, significant differences (Kruskal-Wallis and Bonferroni tests, 95% confidence level) have been detected between the upper and lower Raba River parts under all four scenarios. Moreover, in the majority of the upper sub-basins, the LU changes counterbalanced climate change effects, except for sub-basins

No. 31, 33 and 35). The same pattern has been detected during the summer period with all the upper sub-basins showing positive effects of the LU changes on total sediment yield. Moreover, such an effect has been also observed for all lower sub-basins under the RCP 8.5 long-term scenario (C2.2 + LU), except for sub-basin No. 3. For the autumn months, the Macromodel predictions display again the ability of the LU changes to compensate for climate RCP 4.5 scenarios (C1.1 + LU and C1.2 + LU) for almost the entire Raba River basin. While for the winter period, the majority of sub-basins, in both parts, displayed positive sediment yield changes, signifying that future LU modifications will not counterbalance climate change effects.

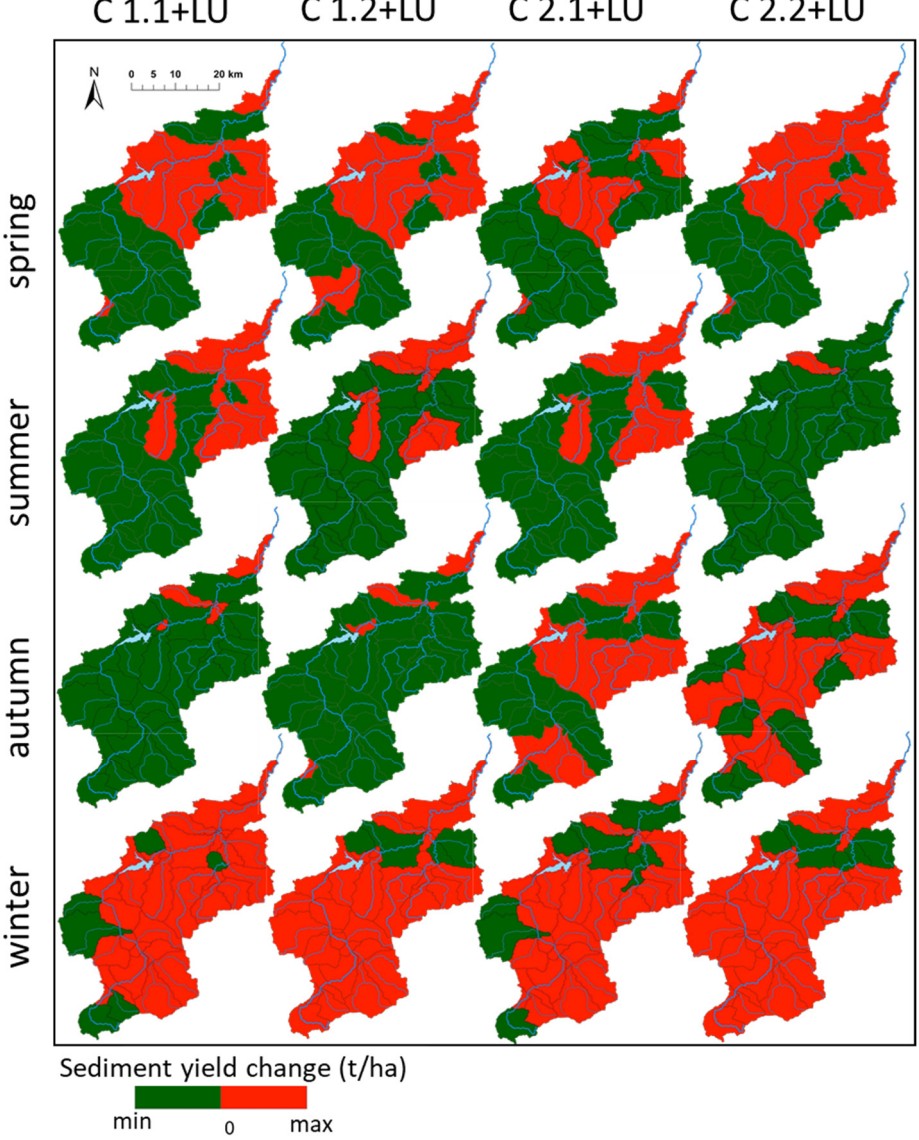

**Figure 11.** Sediment yield changes (t/ha) resulting from superimposing the land use scenario over climate change scenarios: green color depicts sub-basins with the decrease and red color depicts sub-basins with the increase of the sediment yield.

## 4. Discussion

The Raba River basin has been selected as a modeling venue to assess the combined effects of climate and land use change scenarios on sediment yield at the sub-basin level. This basin covers an area elevated from 145 a.s.l. to 1272 a.s.l., and therefore, provides a unique opportunity to study a wide spectrum of land and climate features. The baseline scenario for all the 36 sub-basins revealed

high sediment yields for the upper part of the area, especially for sub-basins No. 29–32 (Figure 8), where slopes exceed 25% (Figure 2a), and agricultural use is about 40% on average in these sub-basins (Figure 3). Although precipitation averaged 1400 mm in this part of the basin, clay soils dominated these sub-basins (Figure 2c), and therefore, are less prone to water erosion and prevented over-excessive leaching of particles. The average sediment yields for these sub-basins reached 1.38 t/ha, while the highest value (1.55 t/ha) was detected at sub-basin No. 17, located in the lower part of the Raba River basin. Although only 10% of its area is covered by high slopes (>25%) and average rainfall reaches 1000 mm, 80% of this area is used for agricultural activities (with 20% covered by potato crops), its soil cover (light dusty clay, light loamy sand, and dusty soil; Figure 2c) appears to be more vulnerable to water erosion. As a combination of land features can account for the record sediment yields, the impact of climate attributes is more visible when seasonal differences are discussed (Figure 8). High levels of precipitation and torrential rainfall during the spring-summer period amplify sediment yield differences between the upper and lower parts of the basin. When a decrease in heavy rain frequency and intensity is observed during autumn, the sediment yield values are considerably reduced, to reach a minimum during winter due to soil freezing, snow cover presence, and low precipitation. Comparing the obtained results to previous analyses related to sediment load and its seasonal variability in the closing profile of the upper Raba catchment [44], it can be concluded that sediment loads and sediment yields do not coincide in particular seasons. As observed previously, the winter sediment loads can reach elevated values, while the winter yields are low (Figure 8). Such differences between phenomena occurring on the catchment surface (yields), and in the river itself (loads), could be associated with the specificity of the Carpathian catchment where deposition processes intensively occur; additionally supported by numerous, but mostly overfilled and improperly managed, anti-erosion structures (Figure 1) [89,90]. They temporarily interrupt the journey of large amounts of sediments washed away to surface waters during the spring and summer precipitation increase. Therefore, during autumn, deposition rapidly increases sediment amounts in such structures. While in winter, along with the frequent occurrence of so-called anomalously wet seasons (AWS) and the local flow increase [72], previously stored sediments begin to move down river. In addition, the erosion of the riverbed is rapidly increasing behind the sediment barriers. It causes an increase in sediment load especially in sub-basins No. 20, 26, 27, 30, 32, and 34.

The spring sediment yield maxima are supposed to be further exacerbated under the adopted climate change scenarios in both time perspectives (Figure 9). Since the predicted temperature changes show a similar increasing pattern for both parts of the basin (Figure 5), this change will be provoked mainly by precipitation increase, especially in the long-term perspective (2071–2100) (by over 25% for RCP 8.5). This is particularly evident in sub-basins No. 25, 30, 32, and 33 where spring sediment yield increased by 0.33 t/ha, compared to the baseline scenario. A similar or even higher precipitation increase has been predicted for the winter season. Thus, the winter sediment yields will require particular attention in this basin. While the forecasts indicate a large increase in rainfall in winter and spring, this increase will be much smaller during summer, and in the case of the long-term forecasts for the Upper Raba, rainfall will decrease even by 3% (Figure 5). Such a summer decrease is characteristic of all mountain areas in this region of Europe, and the higher the analyzed sub-basin is located above sea level, the higher decrease of precipitation should be expected [78]. Therefore, the summer sediment yield decrease is noticeable in many Raba River sub-basins (Figure 9), especially for the long-term RCP 8.5 prediction, with the highest decrease of 62 t/ha in sub-basin No. 31.

In land use scenarios (LU), in which the increase of forest and urban areas was predicted to occur at the expense of arable lands, the maximum sediment yield reduction is expected in spring and summer (Figure 10). The analyses showed that even a 15% increase of the residential medium- and low-density areas has a negligible impact on the size of the sediment yield. Therefore, its reduction will result almost exclusively by the growth of forest areas [91–93]. According to the modeled scenario such changes will be noticeable in the entire Raba River basin, but the most intense changes will concern its upper part (Figure 6). Particularly, sub-basins No. 24–28, 30, 32, 34, and 36 will display a reduction in

spring and summer sediment yields, reaching even 0.37 t/ha (sub-basin No. 34). The predicted 30% increase of forest areas may therefore reduce sediment yields by more than 35% in selected sub-basins, which proves a very good effectiveness of the model's forest soil loss mitigation function.

　　The concurrent discussion of how the combined climate and land use changes impact sediment yields on the basin scale is still not very common in scientific publications. Moreover, due to variability of precipitation, temperature, and land use projections, related to geographic and economic factors, it is difficult to indicate the uniform relationship between these projections and basin response. Nevertheless, the results from very contrasting areas in China [17], the USA [25] or the UK [18] indicate that most likely changes in land use are the key driving force of changing sediment yield. Such a pattern is also clearly visible in the Raba River basin. However, its extent varies greatly in both parts. In its upper, the applied liberal land use forecast, i.e., the increase in forest area by as much as 30%, generally compensated for the sediment yield changes induced by the climate predictions (Figure 11), except for winter. Even in case of the spring months, when sediment production is abundant in sub-basins with high slopes, intense agriculture, and exposed to high precipitation, the yield reduction can reach 10.5–23.2% on average, depending on the scenario. Since the predicted temperature increases are almost uniform among the seasons, the precipitation variations should also be taken into consideration when the effectiveness of land use changes is further discussed. Especially, when the RCP 8.5 long-term summer precipitation decrease, causing over 40% reduction of the sediment yield, is followed by the autumn precipitation increase (Figure 5). This situation results in sediment yield growth which cannot be further attenuated by the forest area increase (sub-basins No. 24, 26, 28, 31–33, and 35). As for the lower part of the Raba River basin the opposite trend could be observed. In the majority of the sub-basins, the land use changes did not compensate for the sediment yields induced by climate change, except for the selected scenarios when only minimal rainfall increases were predicted (summer under C1.2 scenario, and autumn under C2.1, and C2.2; Figure 5). Since, this part of the basin has mainly an agricultural character, with fertile soils and highly prone to erosion (loess), the predicted growth of the forest area share will not be effective to balance out washing of sediments.

　　The most unique situation in the studied basin is, however, created by the winter predictions. The obtained results clearly show that regardless of the chosen scenario, the imposed land use changes will not compensate for the sediment yields induced by the climate predictions in the majority of the sub-basins. Although the lack of plant cover during this period, protecting soil particles against washing out during the vegetation period, is usually accounted for by this phenomenon [16,94,95], meteorological conditions should also be taken into consideration. The predicted temperature increase during the winter months will reduce the time-span of snow accumulation in the higher altitudes and eventually will replace snowfall with rainfall in the entire basin. This situation, combined with the lower infiltration of soils and energy-limited evapotranspiration in low temperatures will increase runoff induced soil leaching [96–98]. The final response of the upper part of the Raba River basin shows a distinct increase of sediment yield, especially in the long-term forecasts, reaching 23–30% on average. While, in the lower part of the basin, such an increase is less pronounced (16–18%). As for the sub-basins displaying that land use changes could still be effective in attenuating sediment yield increase induced by climate change (Figure 10), it should be noticed that both sub-basin groups differ notably in terms of their response to climate change. As visible in Figure 9, the selected upper part sub-basins (No. 21, 23, 24, 34, and 36) display a relatively small sediment yield increase when compared to the rest. Since the sediment yield changes exerted by the land use are uniform for the entire basin, therefore, the impact of climate-induced change will be more decisive. Likewise, in the lower part, the distinctive sub-basins (No. 2, 6–8, 11–12, and 15) are marked by relatively low sediment yield change, and subsequently is even lowered when the land use scenario is applied. Therefore, it can be concluded that while from spring to autumn land use changes have a decisive impact on sediment yields, winter climate changes exert greater importance.

## 5. Conclusions

The example of the Raba River (Carpathian Mts., Poland) demonstrates that even relatively small river basins can display huge variability in terms of land and climate features, which greatly affect sediment yield calculations. Therefore, it seems to be very purposeful to divide the modeled area into smaller units (sub-basins), which could bring even more specific answers to factors controlling the overall basin sediment leaching response. This may especially occur when the river basin can be divided into parts with contrasting features, which here encompass the Carpathian upper part and the sub-montane lower portion of the Raba River basin. The general land use forecasts for this region predict gradual afforestation of the studied basin at the expense of agricultural areas, and a reduced impact of urban areas due to decreasing number of inhabitants. However, the extent of these phenomena is much more visible in the upper part of the basin. Moreover, the temperature and precipitation predictions display noticeable differences, with higher variability of changes for the mountainous part of this area, which is specific to the Carpathian Mts. The response of 36 individual sub-basins to climate scenarios created a mosaic of negative and positive sediment yield changes in comparison to the baseline scenario [88]. The overlapped land use predictions allowed us to indicate those sub-basins where land use changes could balance out sediment yields under climate change predictions, and those where it will not be possible. The general response for the combined scenarios revealed that sediment yields could be altered even by 49% in the selected upper sub-basins during the spring-summer months, while for the lower sub-basins the predicted changes would be less effective, up to 3% on average.

Since these effects are based on the assumed replacement of agricultural areas by forests, it should be noted that it will be a continuously progressive process and not necessarily feasible in all parts of the basin. Moreover, future research should focus on simulations relating the sediment yield to the forest growth and structure. In addition, it must be remembered that replacing agriculture with forest areas is possible in practice, in a limited area. While it is possible on weak and infertile soils on high slopes, this solution will not be possible on fertile lowland catchments due to economic issues. The seasonal changes taken into consideration here also prove that special attention must be paid to the winter months. Under the adopted climate changes the frost and snow cover protecting soils against erosion will become exceptional even in the mountainous part of the basin, while completely disappearing in the sub-mountain part in the long-term perspective. These phenomena will also significantly affect the soil particle transport within the studied basin. It seems to be therefore extremely important to underline that sediment transport modeling, based on the averaged values of temperature, precipitation, and land use changes, can lead to significant errors in the scale of the entire basin. Therefore, it is highly recommended to individualize forecasts at the sub-basin level, as performed here. Moreover, the extent and magnitude of land-use and land management practices vary depending on the needs of local communities, which also prompts a downscaled assessment based on modeling tools. The approach proposed in this current article is the first of its kind in the Carpathian Mts., and must be continued with further analyses for the mountain/sub-mountain basins.

**Author Contributions:** Conceptualization, P.O.-W., E.S. and P.W.; Methodology, P.O.-W.; Software, P.O.-W.; Validation, P.O.-W., E.S. and P.W.; Formal Analysis, E.S.; Investigation, E.S. and P.W.; Resources, P.W.; Data Curation, P.W.; Writing-Original Draft Preparation, P.O.-W., E.S. and P.W.; Writing-Review & Editing, P.O.-W., E.S. and P.W.; Visualization, P.O.-W.; Supervision, E.S. All authors have read and agreed to the published version of the manuscript.

**Funding:** This research was funded by Polish Institute of Meteorology and Water Management-National Research Institute: Research Fund—FBW6 and FBW7 tasks. The described studies used the data "Identification of pressures in water regions and river basins districts. Part II: Development of database of anthropogenic pressures" developed at State Water Holding-Polish Waters.

**Conflicts of Interest:** The authors declare no conflict of interest.

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
