# Peer review of "Do Land Use Changes Balance out Sediment Yields under Climate Change Predictions on the Sub-Basin Scale? The Carpathian Basin as an Example"

_water, doi:10.3390/w12051499_

Round 1
Reviewer 1 Report
Well written manuscript. A few areas where clarity could be improved, such as lines 80-82--word construction is a little difficult to know what is being said/intended.
Lines 38-39: "..assumed that changes...increases in CO2..." No other mention to CO2 in the paper and this statement is not referenced. Remove it--it adds nothing.
Table 3 appears to be a sort of classification for a range of values of each statistical tool and table 4 are the results using this classification. Table 3 requires a fuller and more descriptive caption.
Figures 2 and 3 do not have a reference in the text prior to their appearance.
All tables and figures need to be ensured for clarity, copies here were fuzzy. Also, the text font on tables and figures should, as much as possible, match that used in the text body.
Tables usually are not paste and copy from Excel as these appear. Tables usually have no vertical lines and very few if any horizontal lines. Recheck journal formatting requirements for tables.
Lastly, a profound and significant result stated in lines 527-529 should also be stated as a fundamental result of the research in the Abstract.
Author Response
Thank you for your helpful review. All suggestions have been introduced. Detailed answers to the issues raised are below:
- The sentence in lines 80-82 of the original manuscript has been modified to clarify its meaning (lines 84-86 of the modified manuscript)
- Following the reviewer comment this sentence has been deleted.
- Following the reviewer comment the Table 3 caption has been modified.
- Following one of the Reviewer 2 comments the Figure 1 content has been modified which caused the change in numbering. However, the location of Figure 3 (Figure 2 in the original manuscript) has been changed. A reference to Figure 4 (Figure 3 in the original manuscript) was added before it appeared in the text.
- All the tables in figures have been rechecked following the journal formatting. The text font in tables and figures has also been modified to match the text body.
- Following the reviewer comment the statement originally included in lines 527-529 has been added into the abstract (lines 20-22).
Reviewer 2 Report
The proposed paper is the first of its kind in the Carpathian area and underlines the importance of a study approach carried out on a sub-basin scale to reduce the error in the evaluation of the sediment yield. The authors demonstrated this thesis in the manuscript with detailed data and analysis, however, referring to applications to the USPED method of which, however, they do not report any map and elaboration. Is the data from literature?.
It would also have been useful to insert a geological or lithological map to verify on which type of substrate the differentiated erosion develops in the different sub-basins and to highlight in the text if locally there are hydraulic works or anthropic interventions that would limit erosion, except in the case that the the area is not uninhabited and totally devoid of infrastructure and services.
In the introduction they refer to the general problem of soil erosion in arable lands, on the subject of which I suggest considering and citing a specific paper in addition to those already mentioned:
Dimotta A., Lazzari M., Cozzi M., Romano S., 2017 – Soil erosion modelling on arable lands and soil types in Basilicata, southern Italy. O. Gervasi et al. (Eds.): ICCSA 2017, Part V, Lecture Notes in Computer Science LNCS, 10408, pp. 57–72, 2017. Springer
The authors should clarify these points in their work. The paper can be published following the minor revisions indicated above.
Author Response
Thank you for your valuable comments, we appreciate your help in clarifying our manuscript. Detailed answers to the issues raised are below:
- The current paper is based on the Macromodel DNS/SWAT analysis. The SWAT module calculations are based on the USLE model and its modification MUSLE. All the maps and data used for these calculations have been described in the paragraph 2.2. (lines 91-111 of the modified manuscript). The USPED model has not been used in this study.
- To address this comment the content of Figure 1 has been split into two parts. The presence of the anti-erosion structures has been marked on the general map of the basin (modified Figure 1). The maps displaying basin characteristic features and susceptibility to erosion have been enlarged and included in the modified Figure 2. Moreover, Table 1 has been modified to underline the meaning of these features in each sub-basin.
- The specific paper suggested by the reviewer has been added into manuscript (line 31) and the reference list (No. 10 in the modified version).